

# Methodological advances to improve repeatability of SOA generation in environmental chambers

Austin C. Flueckiger and Giuseppe A. Petrucci

The University of Vermont, Department of Chemistry, 82 University Place, Burlington, VT, 05405, U.S.A.

Correspondence to: Giuseppe A. Petrucci (giuseppe.petrucci@uvm.edu)

**Abstract.** Most laboratory atmospheric chamber studies probing the chemical and physical properties of secondary organic aerosol (SOA) perform such experiments with mixing ratios of volatile organic compounds (VOCs) well-above atmospheric relevance ($\gtrsim$50 ppbv). When performing ozonolysis of biogenic VOCs at mixing ratios of atmospheric relevance ($\lesssim$10 ppbv), repeatability of replicate experiments is hindered by the limitations of conventional VOC injection techniques. To overcome these limitations, two novel components (stop/flow and split valves) were embedded in a conventional VOC injection setup, thereby permitting the use of higher VOC volumes of injection to attain low VOC mixing ratios, and the delivery of the VOC to the environmental chamber as a short, discrete pulse for subsequent reaction. Implementation of these novel VOC injection components has resulted in improvements in variability between replicate chamber experiments of up to a factor of 7 with respect to particle number, mass, and size distributions at both high and low VOC mixing ratios (50 and 10 ppbv, respectively). These improvements permit extension of quantitative measurements of SOA formation to VOC mixing ratios at or near atmospheric levels, where new particle formation (NPF) and SOA mass loading are typically within experimental variability.

## 1 Introduction

It is of immediate interest to better understand the chemical and physical nature of atmospheric aerosols, which have potential effects on climate change, air quality, and human health (Xu et al., 2021). Organic aerosols (OAs) account for a large portion of Earth's fine particle mass budget (up to 90%, depending on the ecosystem), and of the total OA fine particle mass, secondary organic aerosol (SOA) accounts for its majority (70 – 90%) (Fischer et al., 2020; Claflin et al., 2018). SOA is aerosol that forms over multiple generations from the gas-phase oxidation of volatile organic compounds (VOCs) emitted from both biogenic and anthropogenic activity (Fischer et al., 2020; Zhang et al., 2018); yet, greater than two-thirds of the approximate 1000 Tg of non-methane VOCs emitted each year stem from *biogenic* sources (Claflin et al., 2018). To understand the dynamic between aerosols and their potential effect on the atmosphere, laboratory studies focused on the measurement and characterization of aerosols need to better reflect the atmosphere. In a typical laboratory setting, the chemical and physical properties of VOC-derived SOA are studied in batch-mode atmospheric chambers. Batch-mode experiments typically involve flexible Teflon chambers of known dimensions in which the user manipulates the interior environment with regards to reagents, and instruments draw air for measurement; duration and instrumental sampling of a batch-mode experiment is limited by the chamber volume and particle-phase wall losses (Krechmer et al., 2020). This is in opposition to continuous-flow chambers, where reactants continuously flow in and air is sampled at the same volume rate to enable a constant chamber volume (Krechmer et al., 2020). Due to methodological and experimental limitations, current understanding of SOA formation and aging has been achieved through laboratory studies conducted at artificially high VOC mixing ratios ($\xi_{VOC}$, ppbv). Results from these studies are then extrapolated in some manner to infer behavior at atmospherically relevant concentrations. Most laboratory studies of SOA generation have been carried out at $\xi_{VOC}$ of approximately 50 ppbv and higher (see examples; Northcross and Jang, 2007; Kundu et al., 2017; Jackson et al., 2017a, b). Even at these elevated $\xi_{VOC}$, however, there tends to be poor repeatability between replicate experiments; this variability increases dramatically at $\xi_{VOC}$ of atmospheric significance ($\lesssim$10 ppbv), often complicating our ability to make quantitative measurements of SOA at these low levels.

Conventionally, VOC injection for laboratory chamber studies is carried out by delivering a known aliquot of pure VOC to a heated three-neck flask. As the VOC volatilizes, it is carried to the chamber by a continuous flow of purified, particle-free air (Iinuma et al., 2004; Jackson et al., 2017b, a; Kundu et al., 2017; Northcross and Jang, 2007; Claflin et al., 2018; Fischer et al., 2020). Variability in these measurements is likely dominated by the small volumes of VOC that must be injected to attain the low $\xi_{VOC}$ and by the time-dependent evaporation of the VOC. For





example, to attain a $\xi_{VOC} = 10$ ppbv in an 8 m$^3$ chamber, a VOC volume of only 0.53 µL is needed. This poses logistical difficulties when using readily available micro-syringes (commonly on the scale of 0.5-10 µL) with typical precisions of about 1% full scale. Furthermore, as the rate of SOA formation, especially the formation of new particles, may depend on the instantaneous mixing ratios of VOC and oxidant in the chamber, the continuous and variable evaporation rate of the VOC can increase experimental variability significantly. To date, the authors are unaware of

any published quantitative study regarding the repeatability of atmospheric chamber experiments and the variability of particle metrics. Herein, we present a newly developed VOC injection setup to significantly reduce the variability, and subsequently, improve the repeatability of atmospheric chamber experiments at all $\xi_{VOC}$.

## 2    Experimental methods

### 2.1    Reagents

All experiments were conducted with α-pinene (Alfa Aesar, 98%, CAS: 7785-70-8) without further purification. Ozone was generated with a commercial unit by passage of dry, particle-free air through a corona discharge (Ozone

Technologies LLC, Model 1KNT, Jersey City, NJ).

### 2.2    Instrumentation and experimental conditions

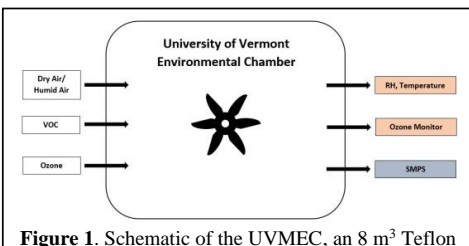

**Figure 1**. Schematic of the UVMEC, an 8 m$^3$ Teflon environmental reaction chamber equipped with two magnetically coupled fans and various sampling/injection ports

For this work, all experiments were performed in batch-mode. Three particle metrics (particle number, mass, and geometric mean diameter) were continuously measured using a Scanning Mobility Particle Sizer (SMPS, TSI Inc., Shoreview, MN), consisting of an Electrostatic Classifier (EC, Model 3080) coupled to a Condensation Particle Counter (CPC, Model 3750). All experiments were performed under ambient temperature (21 ± 2 °C) and atmospheric pressure in the University of Vermont Environmental Chamber (UVMEC, **Figure 1**), an 8 m$^3$ Teflon chamber equipped with two magnetically coupled fans, a VOC injection port (through one fan), reagent injection ports (ozone (O$_3$) and humid air), and analyte sampling ports. A 50 mL, glass three-neck pear flask held at constant temperature in a hot water bath (80 ± 2 °C)

was used for VOC vaporization. A mass flow controller (Sierra Instruments, Model 810S, Monterey, CA) permitted sweeping of the vaporized VOC with dry, particle-free air at a constant flow rate of 2.0 L min$^{-1}$. As described below, a stop/flow valve and split valve were incorporated on the inlet and outlet, respectively (**Figure 2**); depending on the

experimental set, the injection setup had one of four arrangements (referring to stop/flow valve and split valve incorporated, **Table 1**).

### 2.3    Improved injection design

The two novel components regarding the engineering of the VOC injection setup were the stop/flow valve (a ball valve) and the split valve (a tunable, 3-way needle valve) embedded on the inlet and outlet (respectively) of the

injection flask (**Figure 2B**). The purpose of the stop/flow valve was to direct airflow to waste (the room, ambient conditions, no vacuum) prior to injection of the VOC; for experiments utilizing this feature, after VOC injection into the flask, a 45-second heating period was implemented to

allow the VOC to vaporize under no-flow conditions in the flask. After 45 seconds, the stop/flow valve was switched, providing the system with one pulse of dry, particle-free air to carry the fully vaporized VOC in its entirety (compared to conventional techniques that sweep the VOC into the

chamber as it vaporizes) to the environmental chamber. The split valve on the flask exit port permitted greater volumes of VOC to be injected into the flask, yet to still attain low $\xi_{VOC}$ in the chamber by conducting most of the vaporized

**Table 1**. List of the eight α-pinene experimental sets and their varying parameters regarding $\xi_{VOC}$, injection type, and use of stop/flow.

| $\xi_{VOC}$ (ppb) | Direct or Split Injection | Stop/Flow (Y or N) |
|---|---|---|
| 10 | Direct | Y |
| 10 | Direct | N |
| 10 | Split | Y |
| 10 | Split | N |
| 50 | Direct | Y |
| 50 | Direct | N |
| 50 | Split | Y |
| 50 | Split | N |

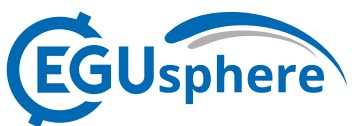

VOC to waste. With conventional injection setup, nanoliter volumes of VOC are needed to establish low $\xi_{VOC}$ in the chamber. Syringes are available commercially with total deliverable volumes as small as 0.5 µL. With typical precisions of 1% of full-scale, the relative error increases as one approaches smaller

volumes (1% at 500 nL to 10% at 50 nL). With the fully optimized injection setup, the ability to inject higher volumes of VOC into the flask while only administering a fraction of the total volume to the chamber allows one to attain atmospherically relevant $\xi_{VOC}$ without

sacrificing precision. For experiments utilizing this feature, the split valve ratio was tunable (injection flow/total flow, $R_{flow}$) and measured explicitly for each experiment to determine the exact volume of VOC needed to attain the desired $\xi_{VOC}$.

**2.4 Methodology**

All ozonolysis experiments were conducted at a $\xi_{O3}$ = 500 ppbv. A total of eight experimental sets were conducted, four at low $\xi_{VOC}$ (10 ppbv) and four at high $\xi_{VOC}$ (50 ppbv), each incorporating one of four possible

VOC injection setups (**Table 1**): direct injection without stop/flow, direct injection with stop/flow, split injection without stop/flow, and split injection with stop/flow. For each, α-pinene was administered to the flask via micro-syringe through a rubber septum. The smallest micro-

syringe necessary to deliver the target volume was selected for each set to minimize reading errors. The desired $\xi_{VOC}$ was determined using Eq. (1):

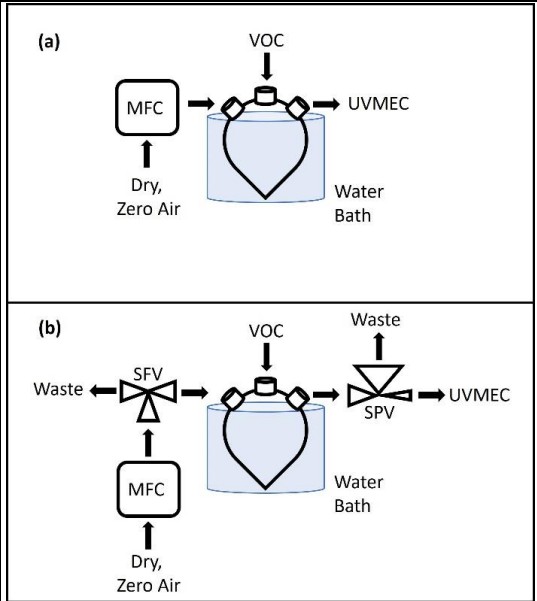

**Figure 2. (a)** Conventional VOC injection setup with continuous airflow through three-neck pear flask directed into the environmental reaction chamber. **(b)** Fully optimized VOC injection setup with the stop/flow valve and split valve incorporated. MFC: Mass Flow Controller; SFV: Stop/Flow Valve; SPV: Split Valve.

$$\xi_{VOC} = \frac{1x10^9\left(\frac{\left(\frac{\rho_{VOC}*V_I}{1x10^3}\right)}{MM_{VOC}}\right)}{C}, \qquad (1)$$

where $\rho_{VOC}$ is the density of the VOC (g mL$^{-1}$), $V_I$ is the volume of VOC injected (µL), $MM_{VOC}$ is the molecular mass of the VOC (g mol$^{-1}$), and C is the constant (332.6243) determined for the UVMEC dimensions. Eq. (1) can be rearranged in terms of $V_I$ to determine the appropriate volume of injection for the desired $\xi_{VOC}$ to form Eq. (2)

$$V_I = \frac{MM_{VOC}\left(\frac{\xi_{VOC}*C}{1x10^6}\right)}{\rho_{VOC}}. \qquad (2)$$

For experimental sets incorporating the stop/flow methodology, the VOC was injected into the flask under static conditions (i.e., all carrier gas flow bypassing the flask and going to waste). After 45 seconds, the stop/flow valve was

140 switched to allow airflow through the flask to the UVMEC, and subsequent gas-phase injection of the VOC. For experimental sets that incorporated the split valve, the flows for both injection-line ($F_I$) and waste-line ($F_W$) were measured with a Gilibrator 2 bubble generator (Sensidyne, LP.; control base P/N 8510190 and flow sensor P/N D800286, Saint Petersburg, FL) and used to determine $R_{flow}$ (Eq. (3))

$$R_{flow} = \frac{F_I}{F_I+F_W}. \qquad (3)$$

The total volume of VOC to inject ($V_T$) could then be determined by Eq. (4)

$$V_T = \frac{\xi_{VOC}}{R_{flow}}, \qquad (4)$$




to attain the desired $\xi_{VOC}$ in the UVMEC. Regardless of a direct injection or split injection, the outlet flow **to the UVMEC** was kept constant at $2.000 \pm 0.013$ L min$^{-1}$.

## 3    Results

### 3.1    NPF, SOA and GMD sensitivity to experimental parameters

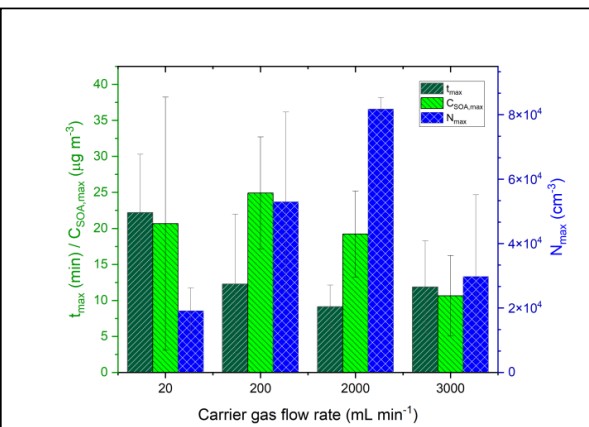

**Figure 3**. NPF of HXL and the variability in time, particle mass, and particle number with respect to carrier gas flow rate (mL min$^{-1}$). The dark green columns represent the total time ($t_{max}$, minutes) from VOC injection to reach maximum particle number ($N_{max}$, cm$^{-3}$, blue). The light green columns represent the maximum SOA mass ($C_{SOA, max}$, μg m$^{-3}$). Each column is color coded to match their respective y-axis.

In any dynamic SOA system, there exists a competition between new particle formation (NPF) and SOA growth through partitioning of lower volatility oxidation products. In a pristine environment (i.e., no existing particles), gas-phase concentrations of oxidation products will grow until either they (1) surpass supersaturation, at which point they may nucleate to a critical diameter that can grow to form new particles, or (2) partition to the newly formed particles, growing them and increasing the SOA mass at a cost to NPF (Donahue et al., 2013; Wiedensohler et al., 2019). One could readily envision that the initial rate of NPF will depend on the instantaneous concentrations of the reactants in the chamber. As the O$_3$ is allowed to equilibrate prior to injection of the VOC, it stands to reason that the rate of injection of the VOC will impact the initial rate of NPF. As mentioned above, in typical VOC injection setups, the VOC is carried to the chamber by a continuous stream of air. However, even for the most volatile of VOCs, evaporation occurs over

a finite time and the vapors are swept to the

chamber as they form. Therefore, the instantaneous chamber VOC concentration will depend on several factors, including the temperature of the injection flask, vapor pressure of the VOC, and carrier gas flow rate. For example, **Figure 3** shows the NPF for the ozonolysis of *cis-3*-hexen-1-ol (HXL) at $\xi_{VOC} = 50$ ppbv and $\xi_{O3} = 400$ ppbv with varying carrier gas flow rates, and continuous flow into the chamber. For low-mid flow rates (20 - 2000 mL min$^{-1}$),

the time to reach maximum SOA particle number ($N_{max}$, cm$^{-3}$) was shortest with the highest flow rate (2000 mL min$^{-1}$), and the greatest $N_{max}$ (~8x10$^4$ cm$^{-3}$) was produced at this flow rate. However, using a stream of air at 3000 mL min$^{-1}$, the flow was strong enough to dilute the VOC injection to the point of suppressed SOA particle and mass formation. Therefore, the 2000 mL min$^{-1}$ flow rate was denoted as a suitable carrier gas flow rate for the experimental sets performed.

### 3.2    Conventional injection setup

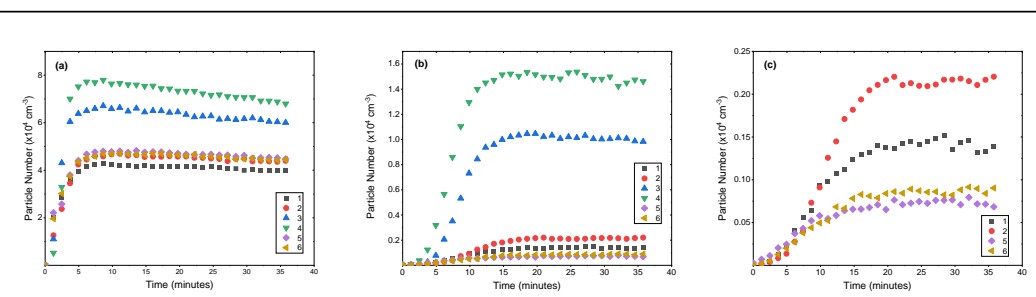

**Figure 4.** Particle number growth for α-pinene-derived SOA with **conventional** injection (direct injection, continuous airflow) for **(a)** $\xi_{VOC} = 50$ ppbv, $\xi_{O3} = 500$ ppbv, n = 6; **(b)** $\xi_{VOC} = 10$ ppbv, $\xi_{O3} = 500$ ppbv, n = 6 (with two statistical outliers to show extreme variability at low $\xi_{VOC}$); and **(c)** the same experimental set as **(b)**, excluding statistical outliers (Runs 3 and 4).





When performing the ozonolysis of α-pinene at $\xi_{VOC}$ = 50 ppbv and $\xi_{O3}$ = 500 ppbv with conventional injection methods (direct injection, continuous airflow, **Figure 4a**), the variability (relative standard deviation, RSD) with respect to $N_{max}$ was measured as 23.3% (which is within the limits that could be inferred from the few literature reports that included multiple trials) (Iinuma et al., 2004; Kenseth et al., 2020). The variability with respect to maximum SOA mass ($C_{SOA, max}$, μg m$^{-3}$) and geometric mean diameter ($GMD_{max}$, nm) was much better, with RSDs of 2.3% and 6.9%, respectively. When extending measurements to lower concentrations at $\xi_{VOC}$ = 10 ppbv and $\xi_{O3}$ = 500 ppbv, the experimental repeatability degraded significantly. When less aerosol is produced at low $\xi_{VOC}$, small deviations in experimental parameters have a larger impact on repeatability. For

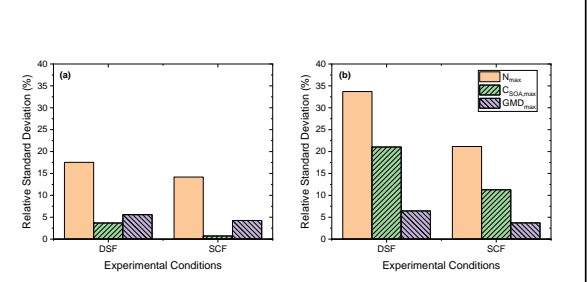

**Figure 5**. Relative standard deviations for three particle metrics for the two intermediate injection setups: direct injection with stop/flow (DSF) and split injection without stop/flow (continuous flow, SCF). **(a)** $\xi_{VOC}$ = 50 ppbv, $\xi_{O3}$ = 500 ppbv; and **(b)** $\xi_{VOC}$ = 10 ppbv, $\xi_{O3}$ = 500 ppbv.

example, six replicate experiments were conducted for each experimental setup; however, while at high $\xi_{VOC}$, all runs were statistically similar, this was not the case for $\xi_{VOC}$ = 10 ppbv. Here, two experiments were statistical outliers (trials 3 and 4) according to a Grubbs test performed at the 90% confidence interval. Both particle growth curves, including and excluding the statistical outliers, are shown (**Figures 4b** and **4c**, respectively) for the purpose of highlighting the difficulty of performing chamber experiments at atmospherically relevant $\xi_{VOC}$ by the conventional VOC injection method. For the statistically valid experimental subset, the RSDs for $N_{max}$, $C_{SOA, max}$, and $GMD_{max}$ were 41.2%, 35.8%, and 4.1%, respectively (**Figure 4c**).

### 3.3 Independent impacts of stop/flow and split flow injection

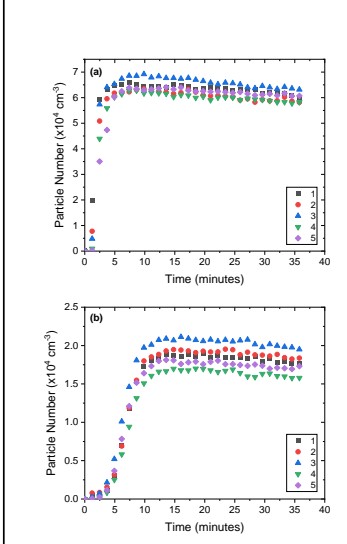

**Figure 6**. Particle number growth curve for α-pinene-derived SOA under **fully optimized** injection setup (split injection, with stop/flow) for **(a)** $\xi_{VOC}$ = 50 ppbv, $\xi_{O3}$ = 500 ppbv, n = 5; **(b)** $\xi_{VOC}$ = 10 ppbv, $\xi_{O3}$ = 500 ppbv, n = 5.

The impact of each injection setup modification on repeatability was studied independently. These were the direct injection with stop/flow (DSF) setup, which tested syringe error (precision error with low volume injections), and the split injection with continuous flow (SCF) setup, which tested the finite vaporization rate (variability of the finite timeframe for sweeping of gas-phase VOC to the chamber). As compared to the conventional injection method, for both high and low $\xi_{VOC}$, the DSF method yielded improvements in repeatability reduction in $N_{max}$ and $GMD_{max}$ for both VOC mixing ratios: RSDs of 17.5% and 5.6%, respectively at $\xi_{VOC}$ = 50 ppbv (**Figure 5a**) and 33.7% and 21.0% at $\xi_{VOC}$ = 10 ppbv (**Figure 5b**). The SCF method, on the other hand, resulted in improvements across **all** particle metrics for both $\xi_{VOC}$ compared to the conventional setup (**Figure 5a and 5b**). For $\xi_{VOC}$ = 50 ppbv, the RSDs for $N_{max}$, $C_{SOA, max}$, and $GMD_{max}$ were 14.2%, 0.7%, and 4.26%, respectively. For $N_{max}$, $C_{SOA, max}$, and $GMD_{max}$ at $\xi_{VOC}$ = 10 ppbv, the RSDs were 21.2%, 11.3%, and 3.7%, respectively. These results suggest that errors in both accurate delivery of the VOC aliquot and control of the vaporization process are essential to improving the repeatability of chamber SOA experiments.

### 3.4 Optimized injection setup

Although the split valve had a greater effect on improved repeatability compared to the stop/flow valve, the use of both components simultaneously (optimized injection setup) improved repeatability up to a factor of ~ 7, compared to the conventional injection setup, for all particle metrics at both $\xi_{VOC}$. For $\xi_{VOC}$ = 50 ppbv (**Figure 6A**), the variability (RSD) for $N_{max}$, $C_{SOA, max}$, and $GMD_{max}$ was 3.5%, 2.3%, and 1.1%, respectively (**Figure 7A**). Of note was the reduction in RSD for $N_{max}$ from 23.3% to 3.5%, an improvement by a factor of 6.7, and $GMD_{max}$ from 6.9% to 1.1%, a factor of



6.3. However, for $C_{SOA, max}$, there was no significant change in RSD from conventional to fully optimized injection setup (2.30 to 2.31%).

Also noteworthy was the reduction in variability in replicate experiments for all particle metrics at atmospherically relevant $\xi_{VOC}$. For $\xi_{VOC}$ = 10 ppbv (**Figure 6B**), the RSDs for $N_{max}$, $C_{SOA, max}$, and $GMD_{max}$ were 7.3%, 7.8%,

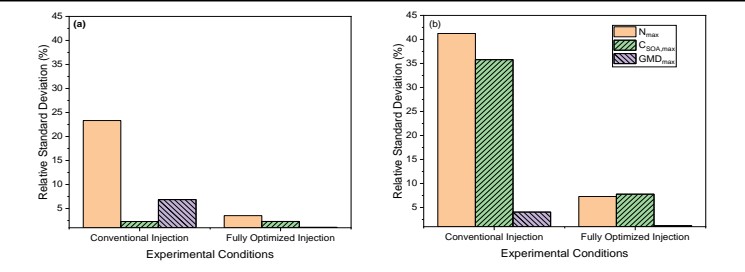

**Figure 7**. The relative standard deviation for three particle metrics with conventional injection setup compared to fully optimized injection setup. **(a)** $\xi_{VOC}$ = 50 ppbv, $\xi_{O3}$ = 500 ppbv; and **(b)** $\xi_{VOC}$ = 10 ppbv, $\xi_{O3}$ = 500 ppbv.

and 1.2%, respectively (**Figure 7B**), which corresponded to improvement factors of 5.6, 4.6, and 3.4, compared to conventional injection methods. Therefore, the use of the stop/flow valve and the split valve, in tandem, had a multi-fold improvement on the repeatability of all particle metrics at high and low $\xi_{VOC}$ for replicate batch-mode experiments (with the exception of $C_{SOA, max}$ at $\xi_{VOC}$ =50 ppbv).

## 4 Conclusions

When performing laboratory experiments to probe the chemical and physical properties of organic aerosol that account for a significant aerosol burden of the Earth's atmosphere, it is crucial that study conditions approximate the natural atmosphere. Yet, performing batch-mode chamber experiments with conventional methodology becomes increasingly difficult as we approach atmospherically relevant reactant mixing ratios. At these lower reactant mixing ratios, the measurement variability introduced by conventional methodology becomes so large as to confuse, if not prohibit, quantitative studies, primarily due to precision errors with VOC injection volumes and VOC vaporization rates.

Herein we have described two innovative improvements in experimental methodology for laboratory chamber studies of SOA formation and growth that provide marked reductions in experimental variability. The addition of a stop/flow valve and a split valve to a conventional batch-mode atmospheric chamber setup is a simple, but effective approach to aid in the mitigation of error in atmospheric measurements by assuring consistency among replicate experimental trials.

*Author contributions*. GP developed the stop/flow valve and AF developed the split valve. AF carried out the experiments and GP supervised the activities. Both authors contributed to the preparation of the manuscript.

*Competing interests*. The authors declare that they have no conflict of interest.

*Acknowledgements*. This material is based upon work supported by the National Science Foundation under Grant No. CHE-1709751.

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
