# Peer review of "Methodological advances to improve repeatability of SOA generation in environmental chambers"

_EGUsphere, 2022_

## Referee Comment (RC2)

The manuscript by Flueckiger and Petrucci developed a new method of adding VOCs for smog chamber study. This method is expected to improve the reproducibility of the chamber experiments. However, the whole work just like the experiment results without deeply scientific. Some parts of this manuscript still need to be supplemented, and it does not meet the requirements of this journal at the current version. The specific comments are listed as follows.

1. Page 1, Section Introduction. The scientific aspects of the need to study SOA formation at low VOCs concentrations should be mentioned in the Section Introduction, such as the fact that VOCs concentrations will affect the distribution of oxidation products and the oxidation state of SOA, as reported by previous studies conducted by Chen et al. (2019, 2022) and Alfarra et al. (2012).

2. Page 2, Section 2.2. My biggest concern is that in order to verify the reliability of the improved method, a sufficient number of parameters of particle properties, including the physical and chemical properties, should be measured. In particular, the chemical properties should be carefully addressed, such as the formation of oxidation products and the oxidation state of SOA.

3. Another of my concerns is whether this new method of adding VOCs affect the formation of gas phase products and the vapor wall loss, which is thought to be crucial to SOA formation, as reported by Zhang et al. (2014).

4. Page 3, Section 2.4. Please check the Eq. (4). $\xi_{VOC}$ should be $V_I$?

5. Page 4, Section 3.1. Why this compound of cis-3-hexen-1-ol (HXL) was chosen instead of the previously mentioned α-pinene?

6. Page 5, Section 3.2. The calculation of RSD is based on the results obtained at the end of the reaction, right? However, this ignored the complex changes in time evolution during the reaction. Thus, more consideration of overall behavior of SOA formation is needed.

Chen, T.; Zhang, P.; Chu, B.; Ma, Q.; Ge, Y.; Liu, J.; He, H., Secondary organic aerosol formation from mixed volatile organic compounds: Effect of RO2 chemistry and precursor concentration. npj Clim Atmos Sci 2022, 5, (1), 95.

Chen, T.; Liu, Y.; Chu, B.; Liu, C.; Liu, J.; Ge, Y.; Ma, Q.; Ma, J.; He, H., Differences of the oxidation process and secondary organic aerosol formation at low and high precursor

concentrations. J Environ Sci 2019, 79, 256-263.

Alfarra, M. R.; Hamilton, J. F.; Wyche, K. P.; Good, N.; Ward, M. W.; Carr, T.; Barley, M. H.; Monks, P. S.; Jenkin, M. E.; Lewis, A. C.; McFiggans, G. B., The effect of photochemical ageing and initial precursor concentration on the composition and hygroscopic properties of beta-caryophyllene secondary organic aerosol. Atmos Chem Phys 2012, 12, (14), 6417-6436.

Zhang, X.; Cappa, C. D.; Jathar, S. H.; McVay, R. C.; Ensberg, J. J.; Kleeman, M. J.; Seinfeld, J. H., Influence of vapor wall loss in laboratory chambers on yields of secondary organic aerosol. Proc Natl Acad Sci USA 2014, 111, (16), 5802-5807.

---

## Author Comment (AC1)

AMT Paper Reviewer #1 Response

**1.** Although it is true that the hypotheses of the studies cited in the introduction do not align with the motivation of our study, the reason for citing these studies was to highlight the typical methodology for VOC injection in batch-mode atmospheric chamber SOA experiments. Unfortunately, to the authors' knowledge, there are no existing reports on the systematic study of variability introduced in chamber measurements due to the method of VOC injection. The authors did not intend for the reader to draw direct correlations with these previous studies, as their motivations were quite different from those of our current report. Additionally, no motivation or parallels are drawn specifically to field measurements of SOA. The prime motivation of our work was to develop a simple, yet effective, means of increasing the repeatability of chamber SOA measurements that could be easily adopted by other researchers. Therefore, by addressing these previous studies, we sought only to demonstrate the typical, or "conventional" procedure for VOC injection, thereby highlighting the main focus of our study, which was the implementation of the stop/flow valve and split valve onto a conventional injection setup to improve repeatability.

**2. a/b.** As there is a continuous flow of dry, particle-free air directed through the flask and into the chamber/split valve throughout the course of the experiment, it is assumed that all of the α-pinene injected is volatized and transferred to the desired location. Therefore, for direct injection experiments, it is assumed that all α-pinene is transferred to the chamber; whereas for split injection experiments, it is assumed that upon entering the split valve, the ratio of α-pinene that is transferred to the chamber vs. waste is proportional to the ratio of air to the chamber vs. air to waste (measured prior to the experiment on the Gilibrator 2 bubble generator). The authors do not have the instrumentation necessary to quantify the mixing rate nor the absolute mixing ratio of VOC in the chamber. Nonetheless, this was not the focus of this work. Despite not having quantified all chamber parameters (such as wall losses of particles and gases, mixing rate, etc.) nor been able to determine the contribution of each to the repeatability of measurements, the results presented here give an upper limit to the repeatability obtainable by making a very simple modification to the VOC injection method. In other words, we know that inclusion of this modification enhances repeatability significantly over conventional methods, but it does not, by any means, represent the best possible. We agree with the reviewer that quantitative knowledge of other chamber parameters that could contribute to measurement variability could improve measurements further.

**2. c.** For this work, the ozone was injected and allowed ample time to mix (10 – 15 minutes) prior to the injection of α-pinene. Although we do not report the calculations of gas partitioning to the chamber walls, we use the particle number concentration, SOA mass, and particle geometric mean diameter as a metric to demonstrate *repeatability*. Specifically, for the fully optimized injection setup, since the variation between replicate experiments (relative standard deviation, RSD) was significantly lowered for both 50 ppbv α-pinene (RSD for $N_{max}$, $C_{SOA, max}$, and $GMD_{max}$ were all ≤ 3.5 %) and 10 ppbv α-pinene (RSD for $N_{max}$, $C_{SOA, max}$, and $GMD_{max}$ were all ≤ 7.8 %), it can be assumed that the contribution of other variables, like VOC/$O_3$ mixing and gas-phase VOC partitioning to the chamber walls, to repeatability enhancements would be less significant. Again, the motivation of the current work was not to fully characterize the chamber, but rather to enhance the repeatability of common measurements made in SOA experiments (that is, particle number density, geometric mean diameter and mass loading).

**2. d.** The authors agree that the absence of seed particles may not represent atmospheric conditions in some cases, but in others, for example the pristine environment, seed-less conditions are wholly appropriate. In any case, seed-less experiments were used for several reasons, including the assumption that they would present the worst-case scenario for repeatability of SOA production specifically for the reasons outlined by the reviewer.

**3. a.** All experiments were performed under ambient temperature (21 +/- 2 C) and atmospheric pressure in the UVMEC (8 $m^3$ Teflon chamber). For all experiments, the relative humidity was 0.0 % (as measured by a Vaisala TMH130 probe).

**3. b.** Due to extensive flushing with dry, particle-free air between replicate experiments (overnight flushing or two hours between same-day experiments), the background sampling of the UVMEC showed no particles present in the atmospheric chamber prior to the injection of the VOC for most experiments. At the very least, background particle number concentration was < 10 particles $cm^{-3}$ before VOC injection.

**3. c.** For the eight experimental sets regarding the repeatability of replicate batch-mode chamber experiments (see **table 1**), α-pinene was the sole VOC used. Prior to this investigation on repeatability, various carrier gas flow rates were tested to compare the particle number and SOA mass yield from the same mixing ratio of *cis*-3-hexen-1-ol (HXL). Based on the results from **figure 3**, a carrier gas flow rate of 2000 mL $min^{-1}$ was implemented as standard procedure for all SOA

experiments performed in the Petrucci Research Laboratory at The University of Vermont. Therefore, when investigating the repeatability of replicate chamber experiments with α-pinene, the justification for specifically using an injection flow rate of 2000 mL min$^{-1}$ stemmed from our previous work with HXL.

**3. d.** To the authors' knowledge, there are no reports in the literature addressing the quantitative repeatability of smog chamber SOA measurements. There are a very limited number of studies that include replicate measurements of SOA produced from α-pinene and $O_3$. For example, Caudillo et al. (2021) performed four replicate α-pinene-derived SOA experiments but assumed a 30% uncertainty based on a study by Dada et al. (2020, which did not use α-pinene as a VOC). Furthermore, Caudillo et al. used a 26.1 m$^3$ stainless steel cylinder chamber and performed these experiments at -30 °C, which cannot be directly compared to our batch-mode atmospheric chamber experiments. In addition, a study conducted by Bonn et al. (2002) reported an RSD of 17% and 18% for maximum particle number and maximum volume concentration, respectively, at mixing ratios of 50 ppbv α-pinene and 110 ppbv $O_3$. These uncertainty values decreased to 9% and 10% with mixing ratios of 1 ppmv α-pinene and 500 ppbv $O_3$. However, these experiments were performed in a 0.57 m$^3$ spherical-glass vessel, and the authors did not report the number of replicate experiments performed. In a study conducted by Jonsson et al. (2006), for five replicate experiments at 19 ppbv α-pinene, the RSD for maximum particle number and maximum SOA mass was 14% and 21%, respectively. Yet, this was conducted in a 140 cm-long Pyrex glass flow tube (volume of 9.5 dm$^3$), which is not directly comparable to our measurements. Lastly, in a study conducted by Yu et al. (2011), replicate measurements (> 3) of 350 ppbv α-pinene (0.5 µL in a 0.216 m$^3$ chamber) had reported RSD values of < 10% for both particle number and SOA mass. This study also included the use of a stop/flow valve, however, the mixing ratio used (approximately 350 ppbv) was significantly higher compared to the mixing ratios used in our study.

**4.** The authors are fully appreciative of the need for high concentrations/mass of reaction products in current chamber experiments due to analytical limitations of current instruments. However, the significance of this work is to demonstrate that simple changes to conventional VOC injection in batch-mode chamber experiments could significantly reduce the variation in typical particle metric measurements for replicate experiments. Although the difference in RSD (across all particle

metrics) from conventional to fully optimized injection setup is more significant for low (10 ppbv) mixing ratios, there is still a dramatic improvement for particle number concentration and geometric mean diameter for high (50 ppbv) mixing ratios. Although the use of low VOC mixing ratio poses other logistical problems (mass spectrometer sampling, understanding intermediate SOA products, etc.), the fundamental takeaway is that the optimization of a conventional VOC injection setup can be applied to laboratory studies that use high concentrations to decrease chamber-experiment variability. Ultimately, current instrumental limits will need to be improved in order to better understand SOA formation at atmospheric VOC mixing ratios.

---

## Author Comment (AC2)

AMT Paper Reviewer #2 Response

1. Although the scope of the paper was to address the repeatability of smog chamber experiments by decreasing variability of particle metric measurements between replicate experiments, we appreciate the reviewer's concern regarding the low VOC concentrations and how they will affect the distribution of oxidation products and oxidation state of the formed SOA. Nonetheless, current work in the authors' laboratory is exactly this, exploring the chemical behavior of SOA at these low concentrations.

2. The reviewer is correct that reliability of the method should include supporting evidence of chemical and physical properties of the SOA formed. Although we demonstrate that repeatability is enhanced for physical properties (and can be implied for chemical properties), a lack of research infrastructure limits the ability to make chemical measurements at low concentrations. Current work in the authors' laboratory is develop a methodology for the chemical analysis of SOA at these low concentrations.

3. By implementing the stop/flow valve and the split valve on a conventional VOC injection setup, we have decreased variability (relative standard deviation, RSD) of replicate SOA experiments in a batch-mode atmospheric chamber. For both 50 ppbv and 10 ppbv α-pinene, the RSDs for $N_{max}$, $C_{SOA, max}$, and $GMD_{max}$ were all $\leq$ 3.5 % and $\leq$ 7.8 %, respectively. Maximum particle number concentration, maximum SOA mass, and maximum geometric mean diameter were measured metrics that were used as a means to demonstrate experimental repeatability. Since the RSDs were significantly lowered compared to the variability for the conventional injection setup, it can be inferred that the RSD for partitioning of gas phase VOC to the chamber walls would simultaneously decrease. By using $N_{max}$, $C_{SOA, max}$, and $GMD_{max}$ as a metric for measuring repeatability, the authors have provided an upper limit to replicate chamber-experiment variability.

4. The authors appreciate the reviewer's correction; the manuscript has been updated accordingly.

5. Prior to this study regarding the improvement of repeatability for replicate batch-mode atmospheric chamber measurements, a study regarding the flow rate of carrier gasses was conducted. Based on the results from **figure 3**, a carrier gas flow rate of 2000 mL min$^{-1}$ was implemented as standard procedure for all experiments performed in the Petrucci Research Laboratory. Although this work was conducted using *cis*-3-hexen-1-ol (HXL) as the precursor VOC, the authors though it necessary to include in this manuscript as justification for the use of a 2000 mL min$^{-1}$ carrier gas flow rate. However, for the main scope of this manuscript, the improved repeatability of replicate batch-mode atmospheric chamber experiments, α-pinene was sole VOC used (**table 1**).

6. The calculation of RSD for each particle metric (particle number concentration, SOA mass, and particle geometric mean diameter) is based on the individual metric's maximum measurement throughout the experiment. For example, the RSD for each experiment's particle number concentration was determined by the maximum particle number ($N_{max}$) throughout the duration of the experiment. The maximum measurement for each particle metric was not always the end of each experiment. In fact, an upcoming report from the authors will specifically address the time evolution of various particle metrics. However, for this manuscript, the maximum measurement for each particle metric was used as a means to demonstrate improved repeatability of replicate experiments.